# Single-Shot Local Injection of Microfragmented Fat Tissue Loaded with Paclitaxel Induces Potent Growth Inhibition of Hepatocellular Carcinoma in Nude Mice

**DOI:** 10.3390/cancers13215505

**Published:** 2021-11-02

**Authors:** Giulio Alessandri, Augusto Pessina, Rita Paroni, Luisa Bercich, Francesca Paino, Michele Dei Cas, Moris Cadei, Arnaldo Caruso, Marco Schiariti, Francesco Restelli, Offer Zeira, Carlo Tremolada, Nazario Portolani

**Affiliations:** 1Cellular Neurobiology Laboratory, Department of Cerebrovascular Diseases, Foundation IRCCS Neurological Institute Carlo Besta, 20133 Milan, Italy; 2Department of Molecular and Translational Medicine, Clinical Microbiology and Virology Section, University of Brescia, 25127 Brescia, Italy; arnaldo.caruso@unibs.it; 3CRC StaMeTec, Department of Biomedical, Surgical and Dental Sciences, University of Milan, 20133 Milan, Italy; augusto.pessina@unimi.it (A.P.); francesca.paino@unimi.it (F.P.); 4Clinical biochemistry and Mass Spectrometry Laboratory, Department of Health Sciences, University of Milan, 20122 Milan, Italy; rita.paroni@unimi.it (R.P.); michele.deicas@unimi.it (M.D.C.); 5Department of Pathology, ASST Spedali Civili of Brescia, 25123 Brescia, Italy; luisa.bercich@spedalicivili.brescia.it; 6Department of Biomedical, Surgical and Dental Sciences, University of Milan, 20133 Milan, Italy; 7Section of Pathological Anatomy DMMT, University of Brescia, 25121 Brescia, Italy; moris.cadei@unibs.it; 8Department of Neurosurgery, Foundation IRCCS Neurological Institute Carlo Besta, 20133 Milan, Italy; marco.schiariti@istituto-besta.it (M.S.); francesco.restelli@unimi.it (F.R.); 9San Michele Veterinary Hospital, 26838 Lodi, Italy; offer@ospedalesanmichele.it; 10Image Regenerative Clinic, Via Mascagni 14, 20122 Milan, Italy; carlo.tremolada@istitutoimage.com; 11Department of Clinical and Experimental Sciences, Surgical Clinic, University of Brescia, 25121 Brescia, Italy; nazario.portolani@unibs.it

**Keywords:** paclitaxel, micro-fragmented fat tissue, hepatocarcinoma, drug delivery, natural scaffold

## Abstract

**Simple Summary:**

Therapeutic approaches to increase localization of chemotherapy at the tumor site, reducing systemic toxicity, are under deep investigation. In previous studies, we have shown that microfragmented adipose tissue (MFAT) may act as a natural scaffold able to deliver anti-cancer drugs. We demonstrated that MFAT and its devitalized counterpart (DMFAT) are able to absorb significant amounts of the chemotherapeutic drug Paclitaxel (PTX), with the ability to kill many different human cancer cell lines in vitro and in vivo, preventing tumor relapse when placed in the surgical area of tumor resection. We demonstrated here for the first time that DMFAT loaded with PTX was also very effective in inhibiting the in vivo growth of hepatocellular carcinoma (HCC) in an advanced stage of progression, suggesting it as a new potent and viable drug-delivery system that may be hypothetically translated to treat inoperable primary tumors in humans.

**Abstract:**

Hepatocellular carcinoma (HCC) is poorly beneficiated by intravenous chemotherapy due to inadequate availability of drugs at the tumor site. We previously demonstrated that human micro-fragmented adipose tissue (MFAT) and its devitalized counterpart (DMFAT) could be effective natural scaffolds to deliver Paclitaxel (PTX) to tumors in both in vitro and in vivo tests, affecting cancer growth relapse. Here we tested the efficacy of DMFAT-PTX in a well-established HCC in nude mice. MFAT-PTX and DMFAT-PTX preparations were tested for anti-cancer activity in 2D and 3D assays using Hep-3B tumor cells. The efficacy of DMFAT-PTX was evaluated after a single-shot subcutaneous injection near a Hep-3B growing tumor by assessing tumor volumes, apoptosis rate, and drug pharmacokinetics in an in vivo model. Potent antiproliferative activity was seen in both in vitro 2D and 3D tests. Mice treated with DMFAT-PTX (10 mg/kg) produced potent Hep-3B growth inhibition with 33% complete tumor regressions. All treated animals experienced tumor ulceration at the site of DMFAT-PTX injection, which healed spontaneously. Lowering the drug concentration (5 mg/kg) prevented the formation of ulcers, maintaining statistically significant efficacy. Histology revealed a higher number of apoptotic cancer cells intratumorally, suggesting prolonged presence of PTX that was confirmed by the pharmacokinetic analysis. DMFAT may be a potent and valid new tool for local chemotherapy of HCC in an advanced stage of progression, also suggesting potential effectiveness in other human primary inoperable cancers.

## 1. Introduction

Hepatocellular carcinoma (HCC) is one of the leading cancers in the world and, despite an improvement in therapeutic options in recent years, the prognosis remains poor [1]. Many factors contribute to the renowned dismal overall survival, such as the low rate of patients that are suitable for radical treatment due to coexisting chronic liver disease and the high rate of recurrence observed after any type of treatment [2,3,4,5]. To date, no scheduled treatment has been clearly defined and accepted to prevent tumor recurrence. For instance, the multi-kinase inhibitors (i.e., Sorafenib) can be used only with strict limitations in a narrow group of patients and, among other therapeutic approaches, the combination of the PD-L1 inhibitor atezolizumab with bevacizumab has recently demonstrated unprecedented results in the treatment of naive patients with an unresectable disease [6]. Among possible therapeutic strategies, systemic therapies are widely implemented and studied, although are typically characterized by low efficacy due to the impossibility of reaching effective concentrations of anticancer drugs specifically at the tumor site due to systemic toxicity. For this reason, the development of strategies to increase chemotherapeutic agent delivery and specific localization at the tumor site would be welcomed. In this regard, the use of magnetic nanoparticles, functionalized with an epithelial growth factor receptor peptide to deliver the anticancer molecule Paclitaxel (PTX), has recently been shown to have some efficacy for in vivo liver cancer therapy [7].

In previous studies, we have shown that adipose tissue (AT), after a process of fat microfragmentation (MFAT) through a commercial device called “Lipogems^®^”, became a natural scaffold able to deliver anti-cancer drugs [8]. More specifically, we found that fresh preparation of MFAT specimens and, surprisingly, even its devitalized MFAT (DMFAT) counterpart, were very effective in adsorbing and releasing a significant amount of the anti-cancer molecule PTX. Both MFAT and DMFAT loaded with PTX (MFAT-PTX; DMFAT-PTX) were able to kill many different human cancer cell lines in vitro when located near tumor cells, with impressive long-lasting anti-cancer activity. In addition, in vivo experiments focusing on DMFAT-PTX activity showed that in nude mice orthotopically transplanted with Neuroblastoma cells and undergoing tumor surgical resection, the local application of DMFAT-PTX blocked or delayed tumor relapse [8]. These results were the first demonstration that DMFAT may represent a very innovative biomaterial that is easy to obtain and able to localize and release anti-cancer molecules at the tumor site.

Based on such observations, here we wanted to verify the efficacy of DMFAT-PTX in inhibiting the growth of an established primary tumor very resistant to chemotherapy, such as HCC. Moreover, we were also interested in evaluating whether a single-shot application of DMFAT-PTX located near a tumor mass could be sufficient to inhibit or delay its growth, quantifying and evaluating the length of the eventual anti-tumor effect. We demonstrated here that both fresh MFAT and DMFAT loaded with PTX were very effective in inhibiting the growth of the HCC tumor cell line Hep-3B, in vitro. In vivo experiments demonstrated for the first time that, while local treatment with the free PTX drug was not effective, a single-shot administration of DMFAT-PTX placed near the HCC tumor mass was strong enough to produce potent growth inhibition.

## 2. Materials and Methods

### 2.1. Sample Collection, Ethics Statements, MFAT and DMFAT Preparation

Samples of Lipoaspirate (LP) were obtained by liposuction of subcutaneous tissue during plastic surgeries, as previously described elsewhere [9,10]. Informed consent was signed by all patients, in accordance with the Declaration of Helsinki. The approval for their use was obtained from the Institutional Ethical Committee of Milan University (n.59/15, C.E. UNIMI, 09.1115).

MFAT specimens were obtained as previously described [9,11]. DMFAT was prepared following a previously published procedure [8,12] consisting of 3 freeze (−20 °C) and thaw (F/T) cycles (see Appendix A).

### 2.2. Chemicals and Reagents

Methanol, isopropanol, acetonitrile, and formic acid (all analytical grade) were supplied by Merck (Darmstadt, Germany). Ammonium formate was purchased from Sigma Aldrich (St. Louis, MO, USA). Water was MilliQ-grade. PTX and PTX-D5 were purchased from Cabru (Arcore, Italy).

### 2.3. Tumor Cell Lines

The in vitro and in vivo anti-cancer activity of PTX loaded into MFAT and DMFAT was evaluated on Hep-3B, an HCC cell line. This cell line was kindly provided by Dr. Valentina Fonsato (Molecular Biotechnology Center, University of Turin, Turin Italy) and was purchased from ATCC. Hep-3B was maintained following ATCC instructions. Briefly, cells were cultured in Eagle’s Minimum Essential Medium (MEM) (Euroclone, UK) supplemented with 10% fetal calf serum FCS (Gibco, Life Technologies, Monza, Italy) and passed weekly at a ratio of 1:5.

### 2.4. Procedure for PTX Priming of MFAT and DMFAT Specimens for In Vitro Experiments

MFAT and DMFAT specimens were primed with PTX following our previous procedure [8]. Briefly, after being washed with Phosphate Buffered Saline (PBS) via centrifugation (200× *g* × 10 min), around 1 mL of MFAT and DMFAT specimens were mixed with different concentrations of PTX (ranging from 0.05 to 4 μg/mL) and prepared fresh from a stock solution of 6 mg/mL (Fresenius Kabi, Italy). Then, they were diluted in MEM +0.2% bovine serum albumin (BSA). After this, samples were shaken and incubated for about 30 min (at 37 °C, 5% CO_2_). At the end of incubation, MFAT-PTX and DMFAT-PTX specimens were washed twice (200× *g* × 10 min) with PBS to remove unbound PTX. Control, untreated MFAT and DMFAT specimens were similarly processed. At this point, both primed and unprimed MFAT and DMFAT specimens were considered ready for studying their biological activity in vitro.

### 2.5. Evaluation of the Activity of MFAT and DMFAT Loaded with PTX on Hep-3B Growth in 2D Assay

MFAT-PTX and DMFAT-PTX preparations were evaluated for anti-cancer activity either by using trans-well inserts (0.4 μm pore size; BD Falcon, NJ, USA) in a co-culture assay with Hep-3B cells or by evaluating the anti-proliferative activity of their conditioned medium (CM) obtained by incubating around 1 mL of MFAT-PTX and DMFAT-PTX specimens cultured in similar volumes of the MEM complete medium for 24 h (37 °C, 5% CO_2_). For the first purpose, around 2 × 10^4^ Hep-3B cells were seeded in wells (24-multiwell plate) and then covered with 700 μL/well of the complete MEM medium and left to adhere for 3 h. Next, different volumes (50, 25, and 12.5 µL) of MFAT-PTX, DMFAT-PTX, or control untreated specimens were placed in the upper compartment of trans-well inserts, covered with 200 μL of the medium, and then placed in the wells with cancer cells. After 72 h of incubation, the direct effect of fat specimens on the tumor cell growth was evaluated either by detaching and counting the Hep-3B in the wells or, in a more limited series of experiments, by staining the adherent Hep-3B cells with 0.25% crystal violet (Sigma Aldrich, MO, USA) and evaluating the optical density of the eluted dye obtained by cell lysing [8].

The anti-cancer activity of CM derived from either MFAT-PTX or DMFAT-PTX specimens was evaluated in a 72-h proliferation assay as previously described [8,13]. The anti-tumor activity of CM from MFAT-PTX and DMFAT-PTX were compared to that of pure PTX and expressed as a PTX-equivalent concentration (p-EC) according to the algorithm p-EC (ng/mL) = IC50 PTX × 100/V50 (µL/well) where IC50 PTX is the concentration of pure PTX producing 50% growth inhibition and V50 is the respective volume of CM that produces the same inhibition.

### 2.6. Histological Analysis of DMFAT–Hep-3B Cells in 3D Constructs

The efficacy of MFAT-PTX or DMFAT-PTX specimens on Hep-3B was also investigated in a 3D assay. Briefly, 50 µL of the control or MFAT-PTX were mixed at 4 °C with 100 µL of Matrigel (BD Biosciences, Franklin Lakes, NJ, USA); Hep-3B (3 and 5 × 10^6^) cells were added and left to jellify for one hour at 37 °C. Then, the complete growth MEM was added to gels and further incubated for 72 h. At the end, the medium was removed, and the gel was processed by immunocytochemical analysis through the cyto-inclusion technique [14] (see Appendix A).

### 2.7. Evaluation of In Vivo Anti-Tumor Activity of DMFAT-PTX

Five-week-old athymic nude-Foxn1nu mice were purchased from Envigo (Envigo, Bresso, Italy) and were housed under pathogen-free conditions. Experiments were reviewed and approved by the licensing and ethical committee of IZSLER (Brescia, Italy) and by the Italian Ministry of Health (Authorization n. 277/2019-PR9 art.31 del D.lgs 26/2014). Note that, for these experiments, only DMFAT specimens were used, because of the advantage given by the possibility to conserve frozen biomaterials until use in mice.

In the first series of experiments, mice (*n* = 6/group) were injected with 5 × 10^6^ Hep-3B cells in the right flank (day 0). The tumors were allowed to grow to an average 0.5–0.7 cm in diameter corresponding to a tumor volume ranging from 65 to 179 mm^3^ (median weight 120 mg) that was calculated using the formula 1/6πd3 [15]. After 10–14 days the mice were randomly subdivided into 4 groups and treated next to the tumor nodule with a single shot of 200 µL saline (control group CTRL), DMFAT (200 µL), DMFAT-PTX 10 mg/kg (200 μg/200 μL), and free PTX drug (200 µg/200 µL saline), respectively. The loading of PTX into just-thawed DMFAT specimens was performed 20–30 min before the injection as described above. After treatments, mice were observed daily; every two days, tumor diameters were measured by caliper. According to the ethical protocol, all the mice were sacrificed when the tumor nodule reached 2.0–2.5 cm in diameter (≥2 g of weight) or, regardless of the diameter, sacrificed on day 60 after the transplant. At this time, only tumor-free mice were followed until 90 days. Animals were euthanized with carbon dioxide inhalation, followed by cervical dislocation.

In the second series of experiments, mice (*n* = 6/group) were similarly injected s.c. with 5 × 10^6^ Hep-3B and treated locally with a half dose of PTX and DMFAT-PTX (100 μg PTX/200 μL) corresponding to 5 mg/kg.

### 2.8. Histology of Hep-3B Tumor Specimens In Vivo

All tumor specimens were formalin fixed and paraffin embedded; sections were stained with Hematoxylin and Eosin (H&E). After the of conventional H&E stains, ten white sections were then obtained from each sample to be processed for immunohistochemical (IHC) staining. Briefly, sections were transferred to glass slides coated with polylysine, deparaffinized in 100% xylene, and rehydrated in graded ethanol. After heat-induced antigen retrieval, endogenous peroxidase was inhibited by incubating tissue sections with 3% hydrogen peroxidase for 15 min at room temperature, whereas specific epitope binding was blocked by incubation for 20 min with 20% human serum. All samples were then processed by the avidine-biotin peroxidase complex method according to the manufacturer’s recommendations (LASB kit Dako Cytomation). In general, IHC was carried out automatically by means of Leica Bond MaxTM technology (Arginase, CK-PAN, CK7, and ANTI-EPATO) and Ventana Bench Mark Ultra (Ki-67): ANTI-EPATO: Antigen retrieval 30′ EDTA, (dil. 1:300 clone OCH1ES Dako-Agilent); ARGINASI: Antigen retrieval 30′ EDTA, (dil. 1:100 Clone SP156 Cell Marque Diapath, KI-67: T.Q. clone 30-9 Roche-Ventana); CK-PAN: Antigen retrieval 5′ Enzyma, (dil. 1:200 clone MNF116 Dako Agilent); CK-7: Antigen retrieval 15′ EDTA, dil. 1:100 clone OV-TL 12/20 Dako Agilent. All stained sections were evaluated on a NIKON ECLIPSE E 600 Microscope equipped with an OLYMPUS DP21 Camera.

### 2.9. Pharmacokinetics (PK) of PTX Released by DMFAT-PTX and Incorporated into Tumor Nodule

PK of PTX in cancer cells when DMFAT-PTX is located nearby was studied in another series of tests. To this end, mice (*n* = 3/group) were injected subcutaneously (sc) with Hep-3B. When the tumor mass was palpable, mice received 200 µL of DMFAT loaded with 100 µg PTX (dose 5 mg/kg), placed sideways to the tumor. Mice were then sacrificed at 2 h, 1, 2, 3, and 7 days after treatments, and blood, tumor nodules, as well as residual subcutaneous DMFAT-PTX tissue were recovered, placed in a tube, and rapidly cooled (−80 °C) until use for the evaluation of PTX content by mass spectrometry. Another group of tumor-free mice received the same sc injection of DMFAT-PTX (at the same dose 5 mg/kg PTX) to investigate the release of PTX in the absence of the cancer nodule.

Biological samples, consisting of whole blood (100 µL), tumor, and fat tissue homogenate (1–5 mg), were supplemented with 25 μL of IS (Paclitaxel D5, PTX-D5 10 μg/mL) and extracted by single-step liquid extraction with methanol/isopropanol (60:40, *v*/*v*). Dry extracts were redissolved with 150 μL of acetonitrile/water (1:1, *v*/*v*), clarified on a 45 µm filter, and 5 μL was injected for LC–MS/MS analysis. All samples were extracted twice. The LC-MS/MS consisted of a Shimadzu UPLC coupled with a Triple TOF 6600 Sciex (Concord, ON, Canada) equipped with a Turbo Spray IonDrive (see Appendix A).

### 2.10. Statistical Analysis

The experiments were performed using MFAT and DMFAT samples obtained from 5 human donors. The in vitro tests were run in triplicate and the reported data are expressed as mean ± standard deviation (SD). Differences between mean values were evaluated according to Student’s *t*-test. A *p*-value < 0.05 was considered to be significant. For in vivo data (six animals in each group), statistical differences were evaluated via the analysis of variance followed by the Tukey–Kramer multiple comparison test. The data distribution and the differences among the SDs were respectively evaluated by the Kolmogorov–Smirnov test and Bartlett’s method. All analyses were performed by GraphPad Prism software, version 4.0 (GraphPad Software Inc., San Diego, CA, USA).

## 3. Results

### 3.1. PTX Displayed Anti-Proliferative Activity on Hep-3B Cell Line In Vitro

Initial experiments were performed to establish the efficacy of PTX to inhibit Hep-3B proliferation. To this end, different concentrations of PTX (ranging from 0.1 ng/mL to 1000 ng/mL) were added to the culture medium. As conducted in previous studies, we calculated the dose of PTX required to reduce 50% (IC50) and 90% (IC90) Hep-3B growth, which was 15 ± 2 and 25 ± 8 ng/mL, respectively (Appendix A) [8,16].

### 3.2. MFAT and DMFAT Specimens Loaded with PTX Exerted a Potent Antitumor Activity on Hep-3B In Vitro

The activity of CM derived from cultured MFAT or DMFAT specimens loaded or not with different concentrations of PTX (0.05 to 4 µg/mL) was investigated. CM from both MFAT-PTX and DMFAT-PTX (0.25 to 4.0 µg PTX) showed similar and strong Hep-3B growth inhibition. However, DMFAT seemed more effective than MFAT, particularly when loaded with a lower dosage of PTX (0.1 µg) (Figure 1A). This trend was confirmed by evaluating p-EC (PTX equivalent concentration). As shown in Figure 1B, both MFAT-PTX-CM and DMFAT-PTX-CM showed dilution-dependent efficacy and p-EC was 12 ± 2 and 18 ± 5 ng/mL, respectively, meaning that DMFAT is able to release more PTX than MFAT after priming with a similar dose of the drug (2 µg/mL). The morphological appearance of Hep-3B cells treated with DMFAT-PTX-CM is shown in Appendix A.

The direct anti-tumor activity of MFAT-PTX and DMFAT-PTX specimens on Hep-3B was also studied using trans-well inserts. Potent inhibition of Hep-3B proliferation was observed and resulted in a correlation with the increasing amount of PTX uploaded (Figure 1C). No significant differences were observed between MFAT-PTX and DMFAT-PTX specimens among the different preparations in terms of efficacy. Control DMFAT and MFAT specimens did not affect Hep-3B proliferation, and besides the reduction in number, Hep-3B cells showing necrotic and apoptotic features were more frequently seen upon being co-cultured with DMFAT-PTX specimens (Appendix A).

Significant inhibition of Hep-3B proliferation was also obtained by reducing the amount of MFAT-PTX and DMFAT-PTX specimens seeded in trans-well inserts (Figure 1D). Even 12.5 µL of DMFAT-PTX (uploaded with 2 µg/mL) was effective enough to produce around 90% Hep-3B growth inhibition. A trend of greater effectiveness of DMFAT-PTX versus MFAT-PTX among AT preparations was noted.

### 3.3. D Matrigel Construct to Study Interaction between Hep-3B Cells and DMFAT Loaded or Not with PTX

To mimic an in vivo situation, 3D constructs obtained by mixing the control and DMFAT-PTX with Hep-3B cells were prepared and then processed for H&E staining (Figure 2). DMFAT showed a typical adipose tissue structure but appeared looser and more disaggregated in the control group (Figure 2A) and in those loaded with PTX 0.5 mg/mL (Figure 2B), if compared to specimens loaded at the maximal dose of PTX 1 mg/mL (Figure 2C). To note, priming of DMFAT with PTX did not affect the homogeneous distribution of Hep-3B cells inside its trabecular/sponge-like structure (Figure 2D–F).

However, the analysis of cancer cell apoptosis (Hoechst 33342 staining) showed very few cells in control DMFAT sections (Figure 2G) compared to all DMFAT-PTX groups where apoptotic cells were significantly increased (around 5- to 10-fold higher than those present in control DMFAT, Figure 2H,I). The apoptotic effect of DMFAT-PTX on Hep-3B was also confirmed by investigating Annexin V expression, which resulted in being significantly higher in DMFAT-PTX compared to the DMFAT control. The apoptotic effect appeared to be mainly dependent on the PTX dose (Figure 2J). We did not observe changes when increasing the number of Hep-3B cells seeded in the 3D constructs (Figure 2K). Similar results were obtained with fresh MFAT-PTX (data not shown).

### 3.4. Anti-Tumor Effect of DMFAT-PTX in an HCC-Established Subcutaneous Growing Tumor

To establish an in vivo HCC model, Hep-3B cells were injected sc in several mice (5 × 10^6^ Hep-3B cells) and when the tumor nodule was palpable, mice were sacrificed and the tumor was removed and analyzed by IHC for the expression of HCC markers. All the tumor nodules analyzed showed intense cellularity composed of large-sized cells, with several atypical mitoses. The in vivo HCC model was confirmed by a diffuse and strong positivity of immunostaining for Anti-Human Hepatocyte-hepar., Arginase, and CK-PAN (Appendix A).

We next evaluated the potential anti-cancer activity of a single-shot DMFAT-PTX administration at the tumor site as described in the Materials and Methods section (Appendix A). Under this schedule of treatments, control mice reached a tumor volume of around 2 cm^3^ in around 30–40 days (Figure 3A). Similar tumor growth behavior was observed in mice treated locally with free drug PTX (Figure 3B) and DMFAT (Figure 3C). In contrast, mice treated with DMFAT-PTX (10mg/kg) showed a significant delay of tumor growth. None of the mice reached 2 cm^3^ in tumor volume even at the maximal time of observation of 60 days. The tumor was 0.904 ± 0.312 cm^3^, which was around half the volume of those in the control, and 33% of mice were even tumor free (Figure 3D). However, using this high PTX dosage, in all treated mice, skin lesions with a necrotic appearance were observed. None of the mice died and such lesions healed spontaneously, without the need for any particular pharmacological treatment, in around 2–3 weeks in all cases (Appendix A). Due to this observation, another group of mice was treated, reducing the concentration of DMFAT-PTX to 5 mg/Kg. At this dose, none of the mice showed skin ulceration at the site of DMFAT-PTX injection and tumor growth was still significantly delayed, reaching around 2 cm^3^ volume after 58 days. At 60 days, 15% of mice were tumor free (Figure 3E). The trend of tumor growth in all groups of mice is shown in Figure 3F. To ascertain that treated mice were tumor-free, the animals that survived after 90 days were sacrificed and the tumor inoculation area was investigated. The histology of the skin tissue confirmed the absence of any residual neoplastic tissue, with disclosure of normal skin with minimal reactive areas of inflammation (Appendix A).

Finally, a histological investigation of the tumor from control and treated mice was performed at the time of mice sacrifice. IHC did not show significant differences in the expression of HCC markers. The CK-PAN-, Arginase-, and Anti-hepato-positive cells were similarly present, and staining was diffused in control as well as PTX- and DMFAT-treated tumors with the exception of DMFAT-PTX-treated mice, where Arginase staining resulted in being more focal (Figure 4). Interestingly, in the tumor of mice injected with control DMFAT, some residual fat tissue was still present at the time of mice sacrifice. This was not seen in the tumor of mice treated with DMFAT-PTX (at both dosages used), suggesting accelerated digestion of the xenotransplanted adipose tissue primed with PTX (Appendix A).

Significant differences were seen by investigating and quantifying Mib-1/Ki67 proliferative markers. Indeed, a significant reduction of Ki67 expression in tumors of mice treated with DMFAT-PTX at both dosages was noted. In particular, in tumors treated at higher PTX dosages, the number of Ki67-positive cells/field was almost five times lower (127 ± 43 versus 654 ± 123 of the control group). A certain reduction of Ki67 expression was also noted in mice treated with the free PTX drug (Figure 5).

### 3.5. PK of PTX Delivered by DMFAT-PTX When Located Nearby Tumor Nodule

The PK of free PTX has been widely studied elsewhere [17,18]. Here we focused on investigating the PK of PTX delivered by DMFAT when located near cancer cells (Figure 6). To this end, a group of mice was injected with Hep-3B cancer cells and when the tumor nodule was formed, DMFAT loaded with a reduced dosage of PTX 5 mg/kg (corresponding to 100 µg/200 µL, to avoid eventual skin ulcerations) was locally injected. A group of control, normal mice (no tumor injected) were similarly treated. In this group of normal mice, the PTX blood concentration at 2 h was 200 ± 34 ng/mL, which decreased to 22.4 ± 5 at 24 h, 17.1 ± 3 at 72 h, and 6.8 ± 2 ng/mL at 168 h. (Figure 6A). The calculation of the apparent volume of distribution at 2 h (Vd2h) gave a very high value (500 mL) that, as expected, confirmed a very low drug localization in the circulatory system [8].

Despite the technical difficulties in recovering all s.c. DMFAT-PTX injected, we also analyzed the residual amount of drug at the site of injection. After 2 h it was 207 ± 14 μg/g (about 41% of the injected amount of 500 µg/g), and this concentration decreased significantly to 101 ± 11 at 24 h, 5.2 ± 0.3 at 72 h, and 1.14 ± 0.1 µg/gr of PTX at 7 days.

The Area Under the Curve (AUC) 0-168 was 38.2 µg/mL/h, and the 7-day local concentration of PTX can be considered to be of pharmacological importance since it is higher than the IC50 of Hep-3B cancer cells (Figure 6B).

In summary, these data are similar to those previously reported using another strain of mice [8]. PTX release kinetics in the presence of the tumor demonstrated significant differences compared to normal mice. After 2 h, PTX in the blood was around 90 ng/mL, which decreased to around 7 ng/mL at 24 h, showing that in the blood of mice bearing tumors, PTX declined more rapidly. In addition, at 72 h, PTX in the blood was below a detectable level (Figure 6C). At 2 h after injection, the DMFAT-PTX implanted near the tumor nodule was, again, very technically difficult to recover completely; however, the concentration of PTX in the residual DMFAT-PTX was significantly lower when compared to those obtained from normal mice, and at 86.58 µg/g (only 17.3% of those initially injected) the nearby tumor nodule was undetectable. In the tumor, PTX detection started after 24 h. At that time, we found that the residual DMFAT and the tumor PTX was 3.35 µg/g and 0.41 µg/g, respectively. At 48 h, in DMFAT, the concentration of PTX was reduced by about 50% (1.81 µg/g), but in the tumor it increased 4 times (1.6 µg/g), thus reaching a concentration of PTX almost equivalent to that of the adjacent DMFAT. In the subsequent analysis performed on days 3 and 7, the PTX concentration in the tumor and in the adjacent DMFAT remained almost equivalent (Figure 6D). The kinetics of the PTX concentration in the tumor and adipose tissue are summarized in Figure 6E,F. By considering the AUC calculated for 7 days (AUC 2-168h) in the tumor mass, we found a value of 0.48 µg/mL/h that is predictive of significant anticancer efficacy because the IC50 of the PTX on Hep-3B cells calculated in vitro was 0.015 µg/g. Of course, the residual DMFAT near the tumor could also contribute to this effect as the AUC 2-168h in DMFAT was very high at 7.12 µg/mL/h. Further details on the procedure for calculating PK of PTX delivered by DMFAT-PTX in vivo are available in the Appendix A.

In summary, these data seem to indicate that tumor nodules initially accelerated the PTX release from DMFAT when located nearby. The equilibrium of the PTX concentration in DMFAT and the tumor was proximally reached at about 48 h, and in the tumor, a concentration of PTX was significantly detectable for at least 7 days; this concentration must be considered of pharmacological importance.

## 4. Discussion

One of the main issues of systemic chemotherapy is its lack of specificity, affecting both cancer cells and normal healthy cells, thus producing undesired side effects [19,20]. Therefore, developing new chemotherapy approaches that may act preferentially at the tumor site is of great interest to improve anti-cancer efficacy and the quality of a cancer patient’s life [21,22]. In this regard, we previously demonstrated that mesenchymal stromal cells (MSCs) can be an optimal tool to deliver anti-cancer drugs such as PTX and Doxorubicine [16,23,24]. Considering the optimal ability of MSCs to act as scaffold cells and the fact that AT is a natural container of MSCs, we hypothesized that MFAT (derivative from liposuction) would work even better in drug delivery, given its histological and structural scaffold-like characteristics [11,12]. The validity of this hypothesis was recently fully demonstrated by our group. We showed the ability of both MFAT and DMFAT in incorporating and releasing PTX; moreover, when DMFAT-PTX was located in the area of tumor resection, it was able to block or delay cancer relapse [8].

In order to expand these previous results, here we asked whether a single-shot DMFAT-PTX administration, located near the tumor mass, would also be effective in inhibiting the growth of a well-established primary HCC tumor in mice. HCC is a tumor for which systemic therapy is considered poorly effective [25,26,27]. The HCC features of the Hep-3B cell line used here were firstly confirmed by histological analysis of the tumor nodule formed after sc cells injection in nude mice. Preliminary in vitro experiments to establish the sensitivity of Hep-3B cells to PTX inhibition revealed a higher IC50 when compared to other tumor cell lines previously investigated [8,16,23]. However, the in vitro activity of MFAT and DMFAT loaded with different concentrations of PTX on Hep-3B showed that both MFAT-PTX and DMFAT-PTX were equally effective in inhibiting Hep-3B proliferation. Moreover, the co-culture and CM were very effective, resulting in being PTX-priming dose dependent. These results were similar to those previously observed using other cancer cell lines [8]. By using a 3D assay that consisted of mixing Hep-3B cells with DMFAT specimens loaded or not with PTX, embedded in a Matrigel matrix to mimic an in vivo situation, we demonstrated that PTX released by DMFAT caused potent cancer cells apoptosis that was directly proportional to the dose of PTX loaded. The apoptotic effect was also confirmed by the analysis of Annexin V expression on Hep-3B and demonstrated that PTX is a cancer cell apoptotic inducer [28].

Although Hep-3B cells have been widely used in vivo to induce HCC tumors [29], we performed a preliminary experiment to establish the dose of Hep-3B necessary to obtain 100% of tumor takes and timing to obtain a palpable tumor of 0.5–0.7 cm in diameter (around 100 mg in weight) in all mice. When the tumors appeared as a clearly palpable mass, DMFAT and DMFAT-PTX specimens were placed as close as possible to cancer nodules. Regarding this aspect, it must be stated that the intra-tumor type of injection would have been the optimal route of treatment, but this application resulted in being technically unfeasible, due to the impossibility of injecting 200 µL (200 mm^3^ in volume) of DMFAT in a smaller tumor volume (100–150 mm^3^). The PTX dose of 10 mg/kg initially used in this study was based on our previous work and was near the maximal drug concentration that can be loaded in 1 mL DMFAT; it was also based on clinical studies in humans [8,30]. DMFAT-PTX treatment produced potent tumor growth inhibition; at 60 days, the tumor weight was around 1 g; in control or in mice treated either with free PTX or DMFAT, the same tumor weight was reached only after 24–31 days and up to 2 g in about 29–36 days. In addition, 33% of DMFAT-PTX treated mice were even tumor free after 90 days post-transplant, as ascertained by histological examination of the skin area.

Notably, the high dose of PTX delivered by DMFAT caused an unwanted effect, producing skin ulceration in all mice with the formation of a necrotic area at the tumor site. However, the area healed spontaneously without pharmacological treatments in any case. Such an aspect was not observed in our previous study where DMFAT-PTX was placed in the area of tumor resection or intraperitoneally injected [8]. We suspect that this unwanted side effect could be caused by the high dosage of PTX delivered and by the efficient drug concentration of DMFAT, particularly when located in the subcutaneous area. Whatever the reason, this result clearly demonstrated the efficacy of DMFAT to restrain the drug mostly at the site of the injection. This conclusion was also supported by the following observations: (1) The local injection of free PTX drug, at similar high dosages, did not induce skin ulcers; (2) reducing the dose of drug uploaded in DMFAT did not induce skin ulcers, although anti-tumor activity was maintained.

Finally, the PK of PTX released by DMFAT loaded with 5 mg/kg of PTX was analyzed in healthy and tumor-bearing mice. In summary, in normal mice, DMFAT retained most of the PTX at the local site of injection, while the blood PTX concentration decreased rapidly showing a PK similar to those previously observed using 6-week-old female immunocompetent BALB/cOlaHsd mice with sc or intraperitoneal DMFAT-PTX injection [8]. Instead, PTX delivered by DMFAT in tumor-bearing mice seemed to have a different PK. We noted that the PTX concentration decreased rapidly in blood; PTX was detectable for only 24 h after subcutaneous DMFAT-PTX injection (in healthy mice, it was for 7 days). Despite the technical difficulties in recovering all DMFAT-PTX injected, particularly in tumor-bearing mice, an accelerated PTX release in the residual peritumoral fat tissue was observed, particularly at an early time (2 h). Only 14% of the total PTX initially injected was recovered (versus 40% in normal mice). In the adjacent tumor, PTX detection was initiated at 24 h with a kinetic that reached drug concentration equilibrium between adipose and tumor tissue after 48 h, remaining detectable in both tissues for 7 days. At this time, the concentration of PTX in the tumor was up to 10-fold higher its IC50 and, therefore, it appeared to be consistent with the anti-tumor effect that we observed when the mice were treated with 5 mg/mL of DMFAT-PTX.

In conclusion, our results, taken together, strongly support our initial hypothesis, and significantly expand previous results proposing MFAT and its derived devitalized DMFAT counterpart as a natural biomaterial able to absorb, transport, and localize chemotherapeutic drugs. We think that this approach possesses significant advantages compared to other scaffolds (natural/synthetic) potentially usable in the oncology field [31,32,33,34]. In fact, beyond the easy availability of the material (the adipose tissue can be easily obtained from any patient through liposuction), the preparation of the MFAT is rapid and is carried out through a closed and sterile system (Lipogems^®^ device) that does not require GMP conditions for clinical use [10], thus also representing an affordable procedure. Furthermore, loading it with the drug is a fast procedure that can be carried out in the same operating room during surgery of a patient with cancer or prepared in advance and stored at −80 °C (DMFAT) before its preferentially autologous use. Hence, in this work, besides further confirmation of the unique biological characteristic of adipose tissue regarding its ability to actively participate in the absorption of drugs and possible “neutralization” of toxicity related to chronic/repeated drug therapies, novel insights were explored regarding its micro-fragmented counterpart, that resulted in being an excellent natural biomaterial capable of absorbing and releasing drugs [35,36,37,38,39]. We showed here that the DMFAT+PTX combination has great pharmacological efficacy. Nevertheless, such preparation could even be theoretically increased by adding other anti-cancer drugs simultaneously in the same “pre-package”, or in different DMFAT preparations of the same patient, opening the possibility of implementation for other types of tumors [10]. From an applicative point of view, in fact, thanks to the low viscosity of the DMFAT-PTX product, its transplantation in the tumor area can be carried out by a simple injection using a syringe with a small needle (22 gauges). Once engrafted at the site of injection, the DMFAT scaffold showed a slow and lasting release of the incorporated pharmacological principle; in fact, we found some residual DMFAT in the tumor even after more than 30 days after injection. This aspect could be very interesting from a therapeutic point of view, as well as from a clinical point of view. There is another aspect to be considered: Most of the available synthetic scaffolds are not completely biodegradable and the accumulation in the host of non-resorbable scaffolds might require a follow-up or, worse, the need to remove them and/or to limit their use for repeated treatments [40]. This aspect, although not completely studied at the present moment, is not necessary when using MFAT and DMFAT scaffolds, due to their complete autologous origin. To note, in our study, also for technical reasons, we performed a single-shot treatment with DMFAT-PTX in mice, although multiple and repeated treatments could be performed hypothetically (also considering the absence of systemic toxicity of the product).

In fact, one of the major limitations of our approach is the insufficient characterization of the MFAT material, and this aspect is due to the heterogeneity and complexity of the cell content of the microfragmented fat, particularly after devitalization through freezing–thawing steps to obtain DMFAT [13]. In addition, other studies are required to better understand the exact mechanism through which MFAT and DMFAT bind PTX (specifically, which lipid/protein components are involved) and how the drug is released, whether as a free drug or through extracellular vesicles [41]. Moreover, it will be fundamental to evaluate the eventual DMFAT-PTX activity in bigger experimental animals (for example pigs) where an intra-tumor injection is feasible as well. Further considering its limitations, this study highlights the great ability of the MFAT scaffold to concentrate a potent anti-cancer drug such as PTX at the tumor site, with significant prolonged anti-tumor activity, with a single local inoculum sufficient to prevent tumor growth and progression in mice for days. In a recent clinical study performed on a dog with mesothelioma treated with canine DMFAT loaded with PTX, potent efficacy in controlling tumor progression was demonstrated. Dog survival was significantly improved and, more importantly, the absence of toxicity and side effects due to DMFAT-PTX treatment. To note, under standard systemic therapy, PTX is particularly toxic in canine species [42]. Whether this approach could also work in humans remains to be determined. However, our approach may hold important new perspectives for cancer therapy. Such a procedure may play a role for those patients affected by inoperable primary tumors, or to prevent relapse/ eliminate local tumor residues after surgery. Such a procedure may also be applied hypothetically to several different types of human cancers, alone or in association with standard chemotherapy protocols. We are confident that, if the results shown in this study could be transferred to the human bedside, malignant cancers of the brain, pancreas, and liver, which currently respond poorly to chemotherapy, may finally have a new therapeutic option.

## 5. Conclusions

This study highlights the great ability of the MFAT scaffold to concentrate a potent anti-cancer drug such as PTX at the tumor site, with a significant prolonged anti-tumor activity with a single local inoculum, suggesting it as a potent and valid new tool for local chemotherapy of HCC in an advanced stage of progression, also suggesting a potential role for the treatment of other human cancers.

## Figures and Tables

**Figure 1 cancers-13-05505-f001:**
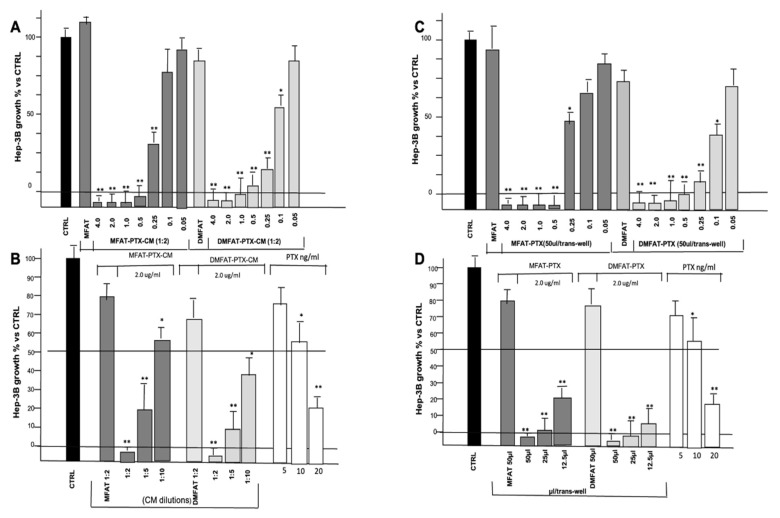
MFAT and DMFAT specimens loaded with different PTX concentrations displayed a dose-dependent anti-cancer activity on human Hep-3B cancer cells in vitro. In the figure, Hep-3B cells’ proliferation activity (expressed as % of control) in the presence of either CM (**A**,**B**) or co-cultured with fat specimens by trans-well insert (**C**,**D**) is reported. Figure 1A reports the anti-proliferative activity of CM (24 h culture) from MFAT and DMFAT loaded with different amounts of PTX (4 to 0.05 μg/mL). Figure 1B shows the CM of DMFAT and MFAT loaded or not with 2 μg/mL PTX and tested at different dilutions. Experiments were run in parallel with free PTX addition to establish p-EC. Figure 1C reports the results of trans-well co-culture of MFAT and DMFAT specimens (50 µL) loaded with different amounts of PTX (4 to 0.05 μg/mL), while Figure 1D shows the anti-proliferative activity of different doses of MFAT and DMFAT specimens loaded with 2 μg/mL PTX, run in parallel, with free PTX addition. Note that CM and specimens recovered from DMFAT primed with PTX at 0.1 μg/mL were more effective than MFAT. Columns in the figures are the means ± SD of two separate experiments, performed in triplicate. *t*-test: * indicate *p* < 0.05; ** *p* < 0.01 versus CTRL MFAT- and DMFAT-derived CM and specimens, respectively.

**Figure 2 cancers-13-05505-f002:**
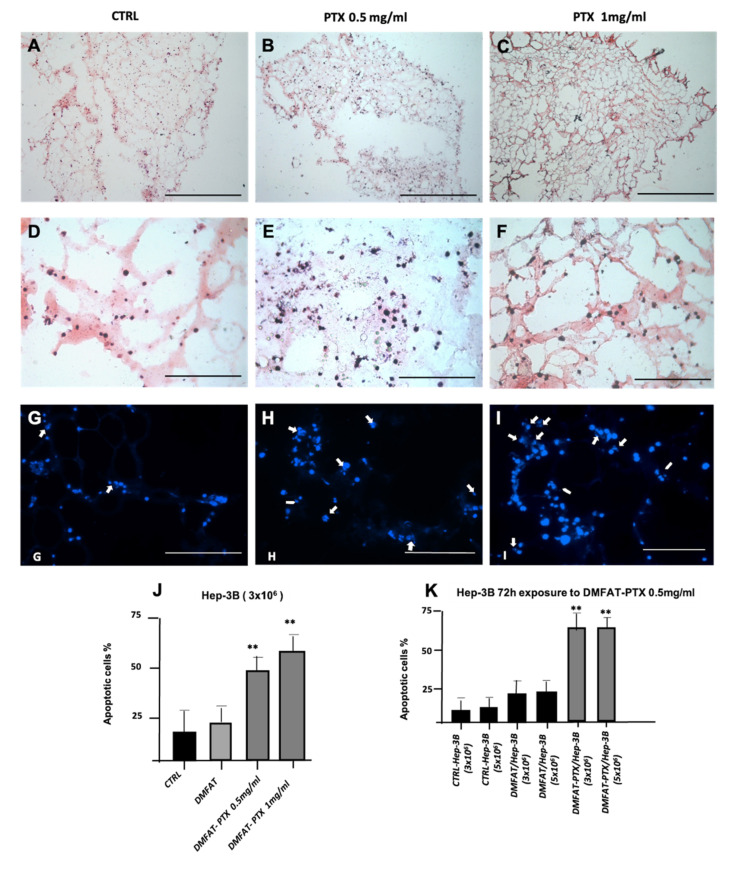
Histological analysis of the 3D constructs mixing Hep-3B cells with control and DMFAT loaded with PTX. H&E staining shows DMFAT with a typical adipose tissue structure (**A**–**C**). DMFAT appeared looser and more disaggregated in control (**A**) and loaded with PTX 0.5 mg/mL groups (**B**) while remaining more compact when loaded with PTX 1 mg/mL (**C**). Hep-3B cells were homogeneously distributed in clusters on DMFAT acting as a matrix scaffold and were more visible at higher magnification (**D**–**F**). The apoptotic effect of DMFAT loaded with PTX on Hep-3B cells detected with Hoechst 33342 staining is shown in G-I ((**G**) control group, (**H**) 0.5 mg/mL, and (**I**) PTX 1mg/mL-treated group)). Chromatin condensation, nuclear fragmentation, and apoptotic bodies are indicated by arrows. Quantification of cell apoptosis was evaluated by Annexin V expression and presented as % of positive cells versus total cells recovered in the constructs incubated with DMFAT uploaded with different PTX concentrations (**J**) or in the presence of different amounts of Hep-3B (**K**). Scale bars = 400 μm (**A**–**C**), 200 μm (**D**–**F**). G-I scale bar of 200 μm. Columns in the figures are the means ± SD of two separate experiments, performed in triplicate. *t*-test: ** indicates *p* < 0.01 versus CTRL DMFAT.

**Figure 3 cancers-13-05505-f003:**
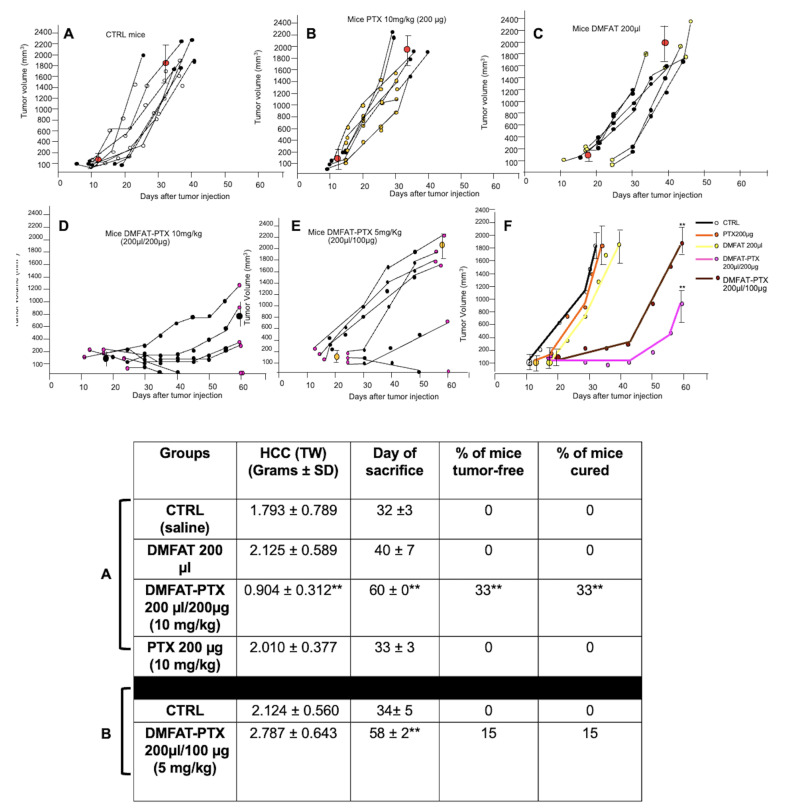
Single-shot local injection of DMFAT-PTX inhibited Hep-3b growth in vivo. Around 5 × 10^6^ Hep-3B were injected sc into nude mice. After 10–12 days, when the tumor was palpable (0.5–0.7 cm diameter around 100–200 mg in weight) mice were injected at the tumor site with saline 200 µL (**A**), PTX 200 µg/200 µL (10 mg/kg) (**B**), control DMFAT 200 µL (**C**), DMFAT-PTX 200 µL/200 µg (**D**), and DMFAT-PTX 200 µL/100 µg (**E**). Tumor volume was calculated by measuring the tumor diameters taken every two days with a caliper using the formula 1/6πd3. In the graphs, the growth of tumor for each individual animal is shown. In (**F**), the average tumor growth of control and treated groups is shown. Note that the single-shot injection of DMFAT-PTX at high dose (200 µg/200 µL) produced a significant delay of tumor growth; at 60 days, 33% of the mice were tumor free (D). Reducing PTX concentration in DMFAT to 100 µg/200 µL still produced significant tumor growth inhibition (F). ** *p* < 0.01 versus saline control or versus DMFAT and free PTX-treated mice. Below, a table reassumes the results of the abovementioned experiments. Note that in the mice receiving DMFAT-PTX 10 mg/kg, tumor weight was significantly reduced at 60 days after injection (** *p* < 0.01); in addition, around 33% of mice were tumor-free and after 90 days were sacrificed and considered cured. Less efficacy was observed in mice treated with DMFAT-PTX loaded with half of the dose of PTX (5 mg/Kg). However, in this group, tumor weight was not significantly reduced compared to CTRL but growth was still significantly delayed (** *p* < 0.01 versus CTRL). In addition, 15% of mice were tumor-free. No significant effect was observed in the group of mice treated with either DMFAT or PTX alone compared to CTRL.

**Figure 4 cancers-13-05505-f004:**
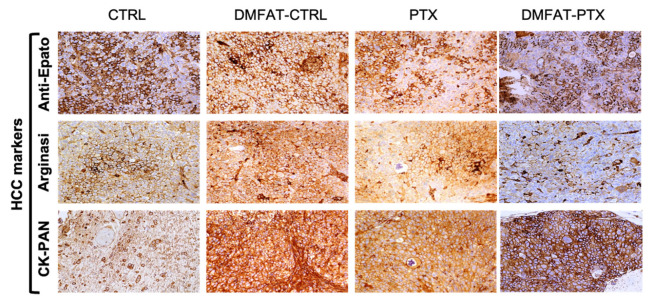
Immunohistochemical staining for typical biomarkers of HCC. DMFAT-CTRL and PTX samples exhibited marked, diffuse positive reactivity for anti-Epato, arginase, and ck-pan markers, similarly to CTRL group. In DMFAT-PTX, at both dosages used, we found less positivity for anti-Epato and Arginase and an increase in CK-Pan expression. Photographs obtained at 10× magnification.

**Figure 5 cancers-13-05505-f005:**
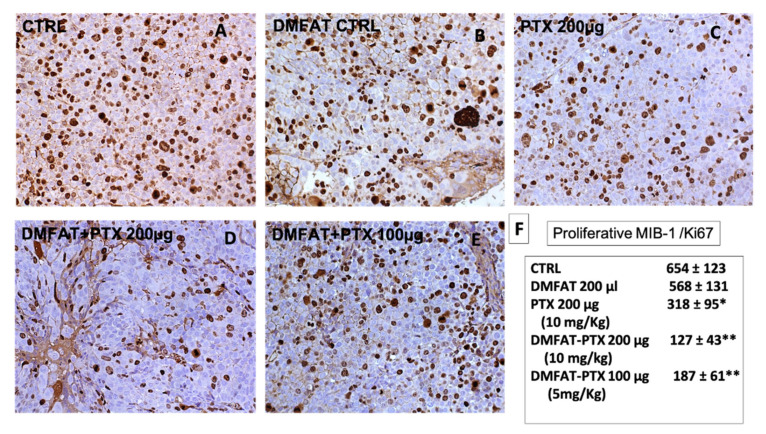
DMFAT-PTX treatment inhibited proliferative marker expression MIB-1/Ki67 in Hep-3B tumor. The nuclear staining of the proliferation marker Ki67 is shown in CTRL group (**A**), in mice treated with DMFAT (**B**), PTX 10 mg/kg (**C**), DMFAT-PTX 10 mg/Kg (**D**), and DMFAT-PTX 5 mg/kg (**E**). A greater positivity for Ki67 expression in CTRL and DMFAT groups was observed. In DMFAT-PTX-treated groups, this proportion was strongly reduced, particularly in DMFAT-PTX 10 mg/Kg. Some significant inhibition was also observed in the free PTX-treated group. (**F**) reports the quantification of the positive Ki67 cells for each group. Numbers represent positive cells counted and represent means ± SD of ten different fields (20× magnification). *t*-test: * indicate *p* < 0.05; ** *p* < 0.01 versus CTRL.

**Figure 6 cancers-13-05505-f006:**
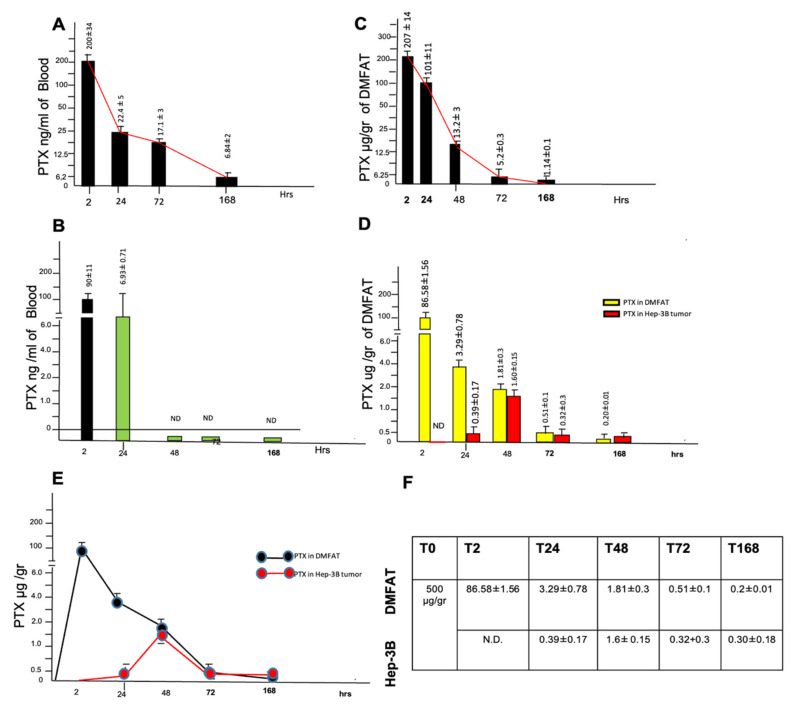
PK of PTX released by DMFAT-PTX in normal and tumor-bearing mice. (**A**,**B**) show the blood concentrations of PTX released by DMFAT-PTX (5 mg/kg) given sc in normal and tumor-bearing mice, detected at 2, 24, 72, and 168 h after injection. As expected, the plasmatic concentration of PTX after sc treatment decreased rapidly but was detectable for 7 days in normal mice (**A**), whereas in tumor-bearing mice, PTX was detectable for only 24 h, suggesting faster removal of the drug from the bloodstream (**B**). (**C**) shows the residual amount of drug in the site of sc injection in normal mice. Note that after 2 h the residual amount of PTX found in situ was about 40% of that injected, whereas after 168 h a significant amount of drug in the residual DMFAT specimens was still detected (up to 1 μg/g), and this must be considered to be of pharmacological importance. In (**D**), the PTX concentration in both residual DMFAT and tumor specimens is shown. After 2 h, in situ PTX was only 17% of the quantity injected while in the tumor it was undetectable. PTX detection in tumor was initiated at 24 h, reaching a concentration equal to adjacent DMFAT at 48 h, and remaining similar until 168 h. At this time, the drug present in the residual DMFAT and in tumor specimens (0.2 μg/g in DMFAT and 0.3 μg/g in tumor) were, however, lower than those recovered in normal mice, but were still higher than the IC50 of the PTX. (**E**) shows the kinetics of PTX detected in both DMFAT and tumor specimens. (**F**) summarizes the values of PTX concentration (expressed as µg/g of tissue) at different times upon DMFAT transplantation. Numbers in the figures are the means ± (up to 1 μg/g) SD.

## Data Availability

Data available on request from the authors.

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
