# Peer review of "Single-Shot Local Injection of Microfragmented Fat Tissue Loaded with Paclitaxel Induces Potent Growth Inhibition of Hepatocellular Carcinoma in Nude Mice"

_cancers, 2021, doi:10.3390/cancers13215505_

Round 1

Reviewer 1 Report

Simple summary

Line 2: “In previous studies we have shown that microfragmented adipose tissue (MFAT) may act as a natural scaffold able to delivery anticancer drugs”.

Change delivery to deliver

Line 3: “We demonstrated that MFAT and its devitalized counterpart (DMFAT) are able to absorb significant amounts of the chemotherapeutic drug Paclitaxel (PTX), being able to kill many different human cancer cell lines in vitro and in vivo prevents tumor relapse when placed in the surgical area of tumor resection”

Change prevents to preventing

Abstract

Line 4: MFAT-PTX and DMFAT-PTX preparations were evaluated for anti-cancer activity in 2D and 3D assays with Hep-3B tumor cells.

“with” should be replaced by against/ using

Line 5: In mice, efficacy of DMFAT-PTX was evaluated after a single–shot subcutaneous injection near a Hep-3B growing tumor by assessing tumor volumes, apoptosis rate and drug pharmacokinetic.

pharmacokinetic should be changed to pharmacokinetics

 Introduction

Paragraph 3

Line 1: Based on such observations, in this study we wanted to verify the efficacy of DMFAT-PTX in inhibiting the growth of a well established primary tumor very resistant to chemotherapy such as HCC.

Lines needs rephrasing.

Materials and methods

2.2. Chemicals and reagents

Ammonium formate was purchased from Sigma Aldrich (St. Louis, MO, USA).

Mark full stop

2.4. Procedure for PTX priming of MFAT and DMFAT specimens for in vitro experiment

Line 2: “Briefly, upon Phosphate Buffered Saline (PBS) washing by centrifugation (200 g × 10 minutes) around 1 ml of MFAT and DMFAT specimens were mixed with different concentration of PTX (ranging from 0.05 to 4 g/ml) and prepared fresh from a stock solution 6 mg/ml and diluted in MEM +0.2% bovine serum albumin (BSA)”.

Check grammar and rephrase

Line 3: Then, samples were shaken and incubated for about 30 minutes (at 37 °C, 5% CO2).

CO2  2 should be in subscript  (Correct all the superscript throughout the manuscript)

2.5. Evaluation of the activity of MFAT and DMFAT loaded with PTX on Hep-3B growth in 2D assay

Line 3: “Briefly, around 2 × 104 Hep-3B cells were seeded in wells (24- multiwell plate) and then covered with 700 μl/well of complete MEM medium and left to adhere for 3 hours”.

2 × 104: Apply superscript to 4

2.6. Histological analysis of DMFAT – Hep-3B cells in 3D constructs

Line 2: Briefly, 50 µl of control or PTX loaded MFAT specimens were mixed at 4°C with 100 µl of Matrigel (BD Biosciences, Franklin Lakes, NJ, USA) where Hep-3B (3 and 5 x 106) cells were added and left to jellify for one hour at 37°C.

Check for grammar and rephrase

Section 2.7 It would be better if “Histology of Hep…..” appears after section 2.8 ”evaluation  of in vivo antitumor…..”

2.7. Histology of Hep-3B tumor specimens in vivoexplain procedure properly

The procedure should be explained in detail.

“All tumor specimens were formalin 10% embedded, included in paraffin; sections with Hematoxylin and Eosin” is incorrect,

Tissue sections must be formalin fixed and paraffin embedded….kindly correct.

2.8. Evaluation of in vivo anti-tumor activity of DMFAT-PTX

Paragraph 2:

Line 2: In the first series of experiments, mice (n=6/group) were injected with 5 × 106 Hep-3B cells in the right flank (day 0).

Check for superscript 5 × 106 (Correct all the superscript throughout the manuscript)

Line 3: The tumors were allowed to grow to an average 0.5/0.7 cm in diameter corresponding to a tumor volume ranging from 65 to 179 mm3 (median weight 120 mg) that were calculated using the formula 1/6πd3 (13).

Correction required: 0.5/0.7 cm, using hypen for refering to range (0.5-0.7 cm)

Superscript correction: mm3

Line 5: After treatments, mice were observed daily; every two days tumor diameters were measured by caliber.

Change calibre to calliper

Last paragraph

In a second series of experiments mice (n=6/ group) were similarly injected s.c. with 5 × 106 Hep-3B and treated locally with half dose of PTX, DMFAT-PTX (100 μg PTX/200 μl) corresponding to 5 mg/kg.

Correct for superscript 5 × 106

2.10. Statistical analysis The experiments were performed using MFAT e DMFAT samples from a total of 5 human donors investigated. Tests were generally run in triplicate and the reported data are expressed as mean ± standard deviation (SD). If necessary, appropriate statistical tests were performed using GraphPad Software (GraphPad Inc., SanDiego, CA, USA). Statistical analysis was also performed with the Statistical Package for Social Science (SPSS version 13, IBM, NY, USA). Statistical differences were evaluated by the analysis of variance followed by Tukey-Kramer multiple comparison test and by the two-tailed, unpaired Student test. p ≤ 0.5 was considered.

Be certain which making statements. Kindly clearly mention the sample size.

“If necessary, appropriate statistical tests were performed using GraphPad Software (GraphPad Inc., SanDiego, CA, USA).” When and where this was used kindly mention.

Mention which parameters are studied by which tests? How is normality of data and homogeneity of variance tested?

Results

3.1

Line 4: “Thus, Hep-3B cells showed a significant resistance to PTX anti-proliferative activity when compared to other cancer cell lines previously tested [6, 14].”

It looks more of result and discussion together. Kindly restrict to result part under this section as discussion section is separate.

3.2

Paragraph 3

Line 2: CM from both MFAT and DMFAT primed with 0.25 to 4.0 µg PTX showed a similar potent Hep-3B growth inhibition.”

Clarify and Rephrase

3.3

Correct 3.33 to 3.3

Paragraph 2

However, the analysis of cancer cell apoptosis detected with Hoechst 33342 staining showed very few cells in control DMFAT sections (Fig. 2G) while in all DMFAT-PTX groups apoptotic cells were significantly increased (Fig. 2H, I) and were around 5 to 10- fold higher than those present in control DMFAT. The apoptotic effect of DMFAT-PTX on Hep-3B was also confirmed by investigating Annexin V expression, that resulted significantly higher in DMFAT-PTX versus DMFAT control. It appeared dependent on PTX dose (Fig. 2J), not being modified by increasing the number of Hep-3B cells seeded in the 3D constructs (Fig. 2K).

Kindly rephrase for easy understanding. Kindly check for grammatical errors too.

Figure 2 Legend

G-I scale bar of 200 mm.

Please check for scale bar measurement in figure G to I

3.4.

Line 1: To establish the HCC nature of Hep-3B cell line, few mice were injected sc with 5 x 106 Hep-3B cells and when the tumor nodule was palpable mice were sacrificed, tumor removed and analyzed by IHC for expression of HCC markers.

The statement should start as “To establish in vivo HCC model, Hep-3B cells were injected……..”

Check for superscript 5 x 106

Line 2: “The hepatocellular nature of the cellularity was confirmed by the diffuse, strong positivity of immunostaining for Anti-Human Hepatocyte -hepar., Arginase and CK-PAN (Suppl. Fig. 4).”

This is not clear to me. Kindly rephrase the statement.

Discussion

Paragraph 1

Line 3: With this aim, we previously demonstrated that mesenchymal stromal cells (MSCs) can be an optimal tool to delivery anti-cancer drugs such as PTX and Doxorubicine [14, 21, 22].

Rephrase

Paragraph 2

Line 5: However, the in vitro activity of MFAT and DMFAT loaded with different concentrations of PTX on Hep-3B. showed that both MFAT and DMFAT loaded with PTX were equally effective in inhibiting Hep-3B proliferation; either the co-culture or CM were very effective and were PTX priming dose dependent.

Remove the highlighted full stop. The line can be split into two for easy reading.

Paragraph 2

Line 2: When the tumors appeared as a clearly palpable mass, DMFAT specimens loaded or not with PTX were placed as close as possible to nearby cancer nodules. In this regard, intra-tumor type of injection, that would have been the optimal route of treatment, was not technically feasible due to the impossibility to inject 200 µl (200 mm3 volume) of DMFAT in a smaller tumor volume. (100-150 mm3).

Remove the highlighted full stop and check for subscripts

Paragraph 5

Line 2: In summary, in normal mice DMFAT retained most of PTX at local site of injection, blood PTX concentration decreased rapidly showing a PK similar to those previously observed but using a different healthy strain of mice [6].

Kindly explain “using a different healthy strain of mice” and specify the strain and study in general detail.

Paragraph 5

Line 10: To note that under standard systemic therapy PTX, in the canine species, is particularly toxic [34].

Kindly rephrase

Author Response

ANSWERS TO THE REVIEWERS

REVIEWER 1

Simple summary

Line 2: “In previous studies we have shown that microfragmented adipose tissue (MFAT) may act as a natural scaffold able to delivery anticancer drugs”.

Change delivery to deliver

ANSWER: Modified as suggested.

Line 3: “We demonstrated that MFAT and its devitalized counterpart (DMFAT) are able to absorb significant amounts of the chemotherapeutic drug Paclitaxel (PTX), being able to kill many different human cancer cell lines in vitro and in vivo prevents tumor relapse when placed in the surgical area of tumor resection”

Change prevents to preventing

 ANSWER: Modified as suggested.

Abstract

Line 4: MFAT-PTX and DMFAT-PTX preparations were evaluated for anti-cancer activity in 2D and 3D assays with Hep-3B tumor cells.

“with” should be replaced by against/ using

ANSWER: Modified as suggested.

Line 5: In mice, efficacy of DMFAT-PTX was evaluated after a single–shot subcutaneous injection near a Hep-3B growing tumor by assessing tumor volumes, apoptosis rate and drug pharmacokinetic.

pharmacokinetic should be changed to pharmacokinetics

ANSWER: Modified as suggested.

Introduction

Paragraph 3

Line 1: Based on such observations, in this study we wanted to verify the efficacy of DMFAT-PTX in inhibiting the growth of a well established primary tumor very resistant to chemotherapy such as HCC.

Lines needs rephrasing.

ANSWER: The sentence was rephrased, as suggested.

Materials and methods

2.2. Chemicals and reagents

Ammonium formate was purchased from Sigma Aldrich (St. Louis, MO, USA).

Mark full stop

ANSWER: Modified as suggested.

2.4. Procedure for PTX priming of MFAT and DMFAT specimens for in vitro experiment

Line 2: “Briefly, upon Phosphate Buffered Saline (PBS) washing by centrifugation (200 g × 10 minutes) around 1 ml of MFAT and DMFAT specimens were mixed with different concentration of PTX (ranging from 0.05 to 4 g/ml) and prepared fresh from a stock solution 6 mg/ml and diluted in MEM +0.2% bovine serum albumin (BSA)”.

Check grammar and rephrase

ANSWER: Checked the grammar and rephrased the sentence, as suggested.

Line 3: Then, samples were shaken and incubated for about 30 minutes (at 37 °C, 5% CO2).

CO2  2 should be in subscript (Correct all the superscript throughout the manuscript)

ANSWER: The subscript was modified throughout the manuscript, as suggested.

2.5. Evaluation of the activity of MFAT and DMFAT loaded with PTX on Hep-3B growth in 2D assay

Line 3: “Briefly, around 2 × 104 Hep-3B cells were seeded in wells (24- multiwell plate) and then covered with 700 μl/well of complete MEM medium and left to adhere for 3 hours”.

2 × 104: Apply superscript to 4

ANSWER: Applied throughout the manuscript, as suggested.

2.6. Histological analysis of DMFAT – Hep-3B cells in 3D constructs

Line 2: Briefly, 50 µl of control or PTX loaded MFAT specimens were mixed at 4°C with 100 µl of Matrigel (BD Biosciences, Franklin Lakes, NJ, USA) where Hep-3B (3 and 5 x 106) cells were added and left to jellify for one hour at 37°C.

Check for grammar and rephrase

ANSWER: Modified as suggested.

Section 2.7 It would be better if “Histology of Hep…..” appears after section 2.8 ”evaluation  of in vivo antitumor…..”

ANSWER: Modified as suggested.

2.7. Histology of Hep-3B tumor specimens in vivo explain procedure properly

The procedure should be explained in detail.

ANSWER: Following the request, a more detailed histological procedure has been added in materials and methods.

“All tumor specimens were formalin 10% embedded, included in paraffin; sections with Hematoxylin and Eosin” is incorrect, tissue sections must be formalin fixed and paraffin embedded….kindly correct.

ANSWER: Modified as suggested.

2.8. Evaluation of in vivo anti-tumor activity of DMFAT-PTX

Paragraph 2:

Line 2: In the first series of experiments, mice (n=6/group) were injected with 5 × 106 Hep-3B cells in the right flank (day 0).

Check for superscript 5 × 106 (Correct all the superscript throughout the manuscript)

ANSWER: Modified as suggested.

Line 3: The tumors were allowed to grow to an average 0.5/0.7 cm in diameter corresponding to a tumor volume ranging from 65 to 179 mm3 (median weight 120 mg) that were calculated using the formula 1/6πd3 (13).

Correction required: 0.5/0.7 cm, using hypen for refering to range (0.5-0.7 cm)

Superscript correction: mm3

ANSWER: The text was modified as suggested.

Line 5: After treatments, mice were observed daily; every two days tumor diameters were measured by caliber.

Change calibre to calliper

ANSWER: Modified as suggested.

Last paragraph

In a second series of experiments mice (n=6/ group) were similarly injected s.c. with 5 × 106 Hep-3B and treated locally with half dose of PTX, DMFAT-PTX (100 μg PTX/200 μl) corresponding to 5 mg/kg.

Correct for superscript 5 × 106

ANSWER: Modified as suggested.

2.10. Statistical analysis

The experiments were performed using MFAT e DMFAT samples from a total of 5 human donors investigated. Tests were generally run in triplicate and the reported data are expressed as mean ± standard deviation (SD). If necessary, appropriate statistical tests were performed using GraphPad Software (GraphPad Inc., SanDiego, CA, USA). Statistical analysis was also performed with the Statistical Package for Social Science (SPSS version 13, IBM, NY, USA). Statistical differences were evaluated by the analysis of variance followed by Tukey-Kramer multiple comparison test and by the two-tailed, unpaired Student test. p ≤ 0.5 was considered.

Be certain which making statements. Kindly clearly mention the sample size.

“If necessary, appropriate statistical tests were performed using GraphPad Software (GraphPad Inc., SanDiego, CA, USA).” When and where this was used kindly mention.

Mention which parameters are studied by which tests? How is normality of data and homogeneity of variance tested?

ANSWER: We re-wrote completely the statistical part, hoping to have better clarified the statistical analysis performed.

Results

3.1

Line 4: “Thus, Hep-3B cells showed a significant resistance to PTX anti-proliferative activity when compared to other cancer cell lines previously tested [6, 14].”

It looks more of result and discussion together. Kindly restrict to result part under this section as discussion section is separate.

ANSWER: Thanks for the suggestion. We modified the text following such suggestion.

3.2

Paragraph 3

Line 2: CM from both MFAT and DMFAT primed with 0.25 to 4.0 µg PTX showed a similar potent Hep-3B growth inhibition.”

Clarify and Rephrase

ANSWER: The sentence was modified following the suggestion.

3.3

Correct 3.33 to 3.3

ANSWER: Modified,as suggested.

Paragraph 2

However, the analysis of cancer cell apoptosis detected with Hoechst 33342 staining showed very few cells in control DMFAT sections (Fig. 2G) while in all DMFAT-PTX groups apoptotic cells were significantly increased (Fig. 2H, I) and were around 5 to 10- fold higher than those present in control DMFAT. The apoptotic effect of DMFAT-PTX on Hep-3B was also confirmed by investigating Annexin V expression, that resulted significantly higher in DMFAT-PTX versus DMFAT control. It appeared dependent on PTX dose (Fig. 2J), not being modified by increasing the number of Hep-3B cells seeded in the 3D constructs (Fig. 2K).

Kindly rephrase for easy understanding. Kindly check for grammatical errors too.

ANSWER: The sentence was rephrased, hoping that now is more easily comprehensible. Thanks for the advice.

Figure 2 Legend

G-I scale bar of 200 mm.

Please check for scale bar measurement in figure G to I

ANSWER: Scale bar was corrected in the legend.

3.4.

Line 1: To establish the HCC nature of Hep-3B cell line, few mice were injected sc with 5 x 106 Hep-3B cells and when the tumor nodule was palpable mice were sacrificed, tumor removed and analyzed by IHC for expression of HCC markers.

The statement should start as “To establish in vivo HCC model, Hep-3B cells were injected……..”

ANSWER: Modified as suggested.

Check for superscript 5 x 106

ANSWER: Modified as suggested.

Line 2: “The hepatocellular nature of the cellularity was confirmed by the diffuse, strong positivity of immunostaining for Anti-Human Hepatocyte -hepar., Arginase and CK-PAN (Suppl. Fig. 4).”

This is not clear to me. Kindly rephrase the statement.

ANSWER: Modified as suggested.

Discussion

Paragraph 1

Line 3: With this aim, we previously demonstrated that mesenchymal stromal cells (MSCs) can be an optimal tool to delivery anti-cancer drugs such as PTX and Doxorubicine [14, 21, 22].

Rephrase

ANSWER: Modified as suggested.

Paragraph 2

Line 5: However, the in vitro activity of MFAT and DMFAT loaded with different concentrations of PTX on Hep-3B. showed that both MFAT and DMFAT loaded with PTX were equally effective in inhibiting Hep-3B proliferation; either the co-culture or CM were very effective and were PTX priming dose dependent.

Remove the highlighted full stop. The line can be split into two for easy reading.

ANSWER: Full stop removed and modified as suggested.

Paragraph 2

Line 2: When the tumors appeared as a clearly palpable mass, DMFAT specimens loaded or not with PTX were placed as close as possible to nearby cancer nodules. In this regard, intra-tumor type of injection, that would have been the optimal route of treatment, was not technically feasible due to the impossibility to inject 200 µl (200 mm3 volume) of DMFAT in a smaller tumor volume. (100-150 mm3).

Remove the highlighted full stop and check for subscripts

ANSWER: Removed the highlighted full stop, subscripts checked, as suggested.

Paragraph 5

Line 2: In summary, in normal mice DMFAT retained most of PTX at local site of injection, blood PTX concentration decreased rapidly showing a PK similar to those previously observed but using a different healthy strain of mice [6].

Kindly explain “using a different healthy strain of mice” and specify the strain and study in general detail.

ANSWER: In the text the strain of mice used previously was added. However, the sentence in the text was referred to previous study (as indicated by the reference 8), where PK and biodistribution experiments were investigated in 6-weeks-old female immunocompetent BALB/cOlaHsd normal mice that were injected with DMFAT-PTX either subcutaneously or intraperitoneally.

Anyway, the text was modified, as suggested.

Paragraph 5

Line 10: To note that under standard systemic therapy PTX, in the canine species, is particularly toxic [34].

Kindly rephrase

ANSWER: Modified as suggested.

Reviewer 2 Report

Dear Editor, thank you so much for inviting me to revise this manuscript about HCC.

This study addresses a current topic.

The manuscript is quite well written and organized. Figures and tables are comprehensive and clear.

The introduction explains in a clear and coherent manner the background of this study.

We suggest the following modifications:

  • Introduction section: although the authors correctly included important papers in this setting, we believe a couple of studies should be cited within the introduction ( PMID: 34126457; PMID: 33820447), only for a matter of consistency. We think it might be useful to introduce the topic of this interesting study.
  • Methods and Statistical Analysis: nothing to add.
  • Discussion section: Very interesting and timely discussion. Of note, the authors should expand the Discussion section, including a more personal perspective to reflect on. For example, they could answer the following questions – in order to facilitate the understanding of this complex topic to readers: what potential does this study hold? What are the knowledge gaps and how do researchers tackle them? How do you see this area unfolding in the next 5 years? We think it would be extremely interesting for the readers.

However, we think the authors should be acknowledged for their work. In fact, they correctly addressed an important topic in HCC, the methods sound good and their discussion is well balanced.

One additional little flaw: the authors could better explain the limitations of their work, in the last part of the Discussion.

We believe this article is suitable for publication in the journal although some revisions are needed. The main strengths of this paper are that it addresses an interesting and very timely question and provides a clear answer, with some limitations.

We suggest the addition of some references for a matter of consistency. Moreover, the authors should better clarify some points.

Author Response

ANSWERS TO THE REVIEWERS

REVIEWER 2

Dear Editor, thank you so much for inviting me to revise this manuscript about HCC.

This study addresses a current topic.

The manuscript is quite well written and organized. Figures and tables are comprehensive and clear.

The introduction explains in a clear and coherent manner the background of this study.

We suggest the following modifications:

Introduction section: although the authors correctly included important papers in this setting, we believe a couple of studies should be cited within the introduction (PMID: 34126457; PMID: 33820447), only for a matter of consistency. We think it might be useful to introduce the topic of this interesting study.

ANSWER: The references suggested by reviewer were added in the introduction section, as requested.

Methods and Statistical Analysis: nothing to add.

ANSWER: Thanks very much.

Discussion section: Very interesting and timely discussion. Of note, the authors should expand the Discussion section, including a more personal perspective to reflect on. For example, they could answer the following questions – in order to facilitate the understanding of this complex topic to readers: what potential does this study hold? What are the knowledge gaps and how do researchers tackle them? How do you see this area unfolding in the next 5 years? We think it would be extremely interesting for the readers.

However, we think the authors should be acknowledged for their work. In fact, they correctly addressed an important topic in HCC, the methods sound good and their discussion is well balanced.

One additional little flaw: the authors could better explain the limitations of their work, in the last part of the Discussion.

ANSWER: We thank the reviewer for the very kind comments to our work. As suggested by reviewer, the discussion section was implemented by adding our opinion about the potential future perspectives of our results. In addition, some other major limits of our work were discussed as well.

We believe this article is suitable for publication in the journal although some revisions are needed. The main strengths of this paper are that it addresses an interesting and very timely question and provides a clear answer, with some limitations.

We suggest the addition of some references for a matter of consistency. Moreover, the authors should better clarify some points.

Round 2

Reviewer 2 Report

The authors modified the paper according to our suggestions.

We recommend Acceptance in its current form.